# Damage Evolution of Polypropylene–Basalt Hybrid Fiber Ceramsite Concrete under Chloride Erosion and Dry–Wet Cycle

**DOI:** 10.3390/polym15204179

**Published:** 2023-10-21

**Authors:** Hongbing Zhu, Siyu Wen, Xiu Li, Yahan Li, Zhenghao Fu

**Affiliations:** 1School of Urban Construction, Wuhan University of Science and Technology, Wuhan 430065, China; 18970520527@163.com (S.W.); 593352366@163.com (Y.L.); whfzh1998@163.com (Z.F.); 2Institute of High Performance Engineering Structure, Wuhan University of Science and Technology, Wuhan 430065, China; 3Hubei Provincial Engineering Research Center of Urban Regeneration, Wuhan University of Science and Technology, Wuhan 430065, China; 4School of Transportation Engineering, Wuhan Technical College of Communications, Wuhan 430065, China; hubeilixiu@163.com

**Keywords:** PBHFCC, dry–wet cycle, chloride erosion, durability damage test, durability damage evolution equation

## Abstract

To investigate the influence of polypropylene–basalt hybrid fibers (PBHFCC) on the durability of ceramsite concrete, this study determined the appearance change, mass loss rate, relative dynamic elastic modulus, compressive strength and splitting tensile strength of ceramsite concrete with four kinds of hybrid fibers volume admixture under chloride erosion and dry–wet cycles. The results reveal that under this effect, the apparent damage of each group of specimens increased with the growth of the erosion time. The quality, compressive strength and splitting tensile strength of the specimens all increased gradually during the erosion age period of the first 72 d and gradually decreased after 72 d. The relative dynamic elastic modulus was similarly mutated in 48 d. When the hybrid fiber content of the specimens is 0.15 vol %, the enhancement effect of ceramsite concrete is better than that of the other three amounts. The relative dynamic elastic modulus value is used as a damage variable to establish the damage equation, and the damage evolution equation of PBHFCC considering the volume of hybrid fiber under chloride erosion and dry–wet cycle is derived. The conclusions can be used as a reference for the durability design and construction of PBHFCC.

## 1. Introduction

Ceramsite concrete is a kind of light aggregate concrete made of ceramsite granule, ceramsite sand and cementitious materials, which has the advantages of high strength, lightweight, low elastic modulus and good sound absorption and noise reduction performance [1]. In order to make up for the defects of ceramsite concrete, fiber materials can be added into ceramsite concrete under the erosion of harsh environment, which can effectively improve the strength [2], toughness [3], crack resistance [4] and freeze–thaw resistance [5] of ceramsite concrete.

With the development of fiber-reinforced concrete, hybrid fiber-reinforced concrete has gradually become a research focus. Hybrid fiber-reinforced concrete is concrete in which two or more fiber-reinforcing materials are added to the cement matrix in a reasonable proportion, which can play the advantages of a single fiber, but also reflects the synergistic effect between the fibers. Therefore, hybrid fiber-reinforced concrete is a new type of composite material with synergistic effect [6,7,8], which can significantly increase or improve the mechanical and working properties of single-fiber or ordinary concrete [9]. When selecting the type of fibers to be blended, rigid and flexible fibers are generally selected for blending. Among them, steel fibers and basalt fibers are more often used as rigid fibers. Although steel fibers can effectively improve the tensile strength and fracture properties of concrete and prevent the expansion of cracks within the concrete, steel fiber has a large weight, is prone to rust and is expensive, so it is not suitable for coastal environments and salt lake areas. Basalt fiber has high tensile strength [10], good high-temperature resistance and stable chemical properties [11], which is a new type of green and environmentally friendly material. Polypropylene fibers belong to flexible fibers with good ductility and are very stable in acid-base environments. The addition of polypropylene fibers to concrete can enhance the crack resistance, toughness and ductility of concrete [12], resulting in denser concrete, lower internal porosity and fewer microcracks on the surface. Polypropylene fiber can effectively alleviate the plastic cracking and drying shrinkage of concrete in the early stage [13], and significantly improve the durability of concrete, so it is suitable to be used as a flexible fiber in hybrid fibers.

In recent years, the research and use of basalt fiber and polypropylene fiber has gradually increased. Jiang et al. [14] state that the basalt fiber content increment results in an increase in the splitting tensile strength by about 15–25%. In addition, it is observed that the splitting tensile strength for the concrete with a fiber length of 22 mm provides better results than that of concrete produced with a basalt fiber length of 12 mm. Zeng et al. [15] investigated the effect of basalt fibers and polypropylene fibers on the mechanical properties of lightweight aggregate concrete. The results illustrated that the incorporation of basalt fiber and polypropylene fiber in concrete had relatively little effect on the compressive strength of concrete, but the effect on the tensile strength, bending strength and shear strength of splitting was obvious. Yao et al. [16] reported the effects of different temperatures and different fiber content on the compressive and flexural strength of basalt–polypropylene hybrid fiber-reinforced mortar and further analyzed the properties of fiber-reinforced cement-based materials through the mechanism of action and microscopic characterization of fibers. The strengthening and failure mechanisms of fiber-reinforced mortars at different temperatures were elucidated. Fu et al. [17] systematically studied the mechanical properties of mixed basalt-fiber- and polypropylene-fiber-reinforced concrete with different matrix strengths and analyzed the mechanism of action of basalt fiber and polypropylene fiber. Moreover, a prediction model for the mechanical properties of hybrid fiber-reinforced concrete was established, and its effectiveness was verified. It can be seen that the influence of basalt fiber and polypropylene fiber on the mechanical properties of ceramsite concrete is a research hotspot. However, the research and application of PBHFCC are not substantial, especially the research on the durability of PBHFCC, which is still in the exploratory stage.

Most of the existing studies mainly focus on the enhancement law of hybrid fibers on the mechanical properties of concrete under normal conditions. When located in the range of frequent water-level fluctuations such as tidal and wave splash zones in marine or salt lake areas, concrete is subjected to both dry–wet cycles and chloride erosion [18,19,20]. After the chloride salt invades the concrete, free chloride ions are distributed in the pores of the concrete [21], resulting in corrosion of the steel bar [22]. The physically adsorbed bound chloride ions are also easily converted into free chloride ions when subjected to external forces, further aggravating the rust of the steel bar. Chloride ions in chemically bound states can react with C3A to form Friedel’s salt [23], which has a great influence on the mechanical properties of concrete such as compressive strength [24]. When concrete is subjected to dry–wet cycles, it will be in a state of continuous water absorption expansion and dehydration contraction. The generation and development of microcracks inside concrete are also further accelerated. The surface hole structure of the concrete specimens is destroyed by the dry–cycle, and the number of pores with a pore size > 50 nm increases. As a result, the concentration of chloride ions entering the concrete is increased [20,25,26,27]. In general, chlorine salts and dry–wet cycles can accelerate the durability damage of concrete, which solicits great concern from the engineering community.

Therefore, it is of great significance to study the durability of PBHFCC. Relevant scholars have studied the durability of different types of fiber-reinforced concrete under chloride erosion and dry–wet cycle. Chen et al. [28] studied the chloride-ion penetration resistance of different types of slag containing fiber-reinforced concrete under dry–wet cycle. The results showed that the apparent diffusion coefficient of chloride ions decreases exponentially with the increase in exposure time. The resistance to chloride-ion penetration of microfiber concrete was mainly related to the fiber volume fraction. The higher the volume fraction of microfibers, the more obvious the diffusion of chloride ions in concrete. The lower volume fraction of ultrafine fibers will enhance the ability of concrete to resist chloride-ion penetration. Algin et al. [29] added 0 vol %, 0.1 vol %, 0.3 vol % and 0.5 vol % basalt fiber into self-compacting concrete and carried out compressive strength, flexural strength, splitting tensile strength, rapid chloride-ion permeability and water permeability tests. The results showed that the use of basalt fiber improves the mechanical properties of self-compacting concrete but reduces the chloride permeability of self-compacting concrete. The compressive strength of concrete with 0.1 vol % fiber content is the highest, which is increased by about 9.5%. In contrast, Guo et al. [30] configured 0 vol %, 0.15 vol %, 0.30 vol %, 0.45 vol % and 0.60 vol % basalt-fiber-reinforced concrete and tested the compressive strength and chloride-ion diffusion coefficient of the concrete. The results showed that with the increase in basalt fiber volume content, the compressive strength of the mixture increased first and then decreased, and reached the maximum when the basalt fiber is 0.15 vol %. The chloride-ion penetration resistance of concrete was weakened by the addition of basalt fiber. In view of the ability of fiber to resist the chloride-ion penetration of concrete, the conclusions drawn by different scholars are not consistent. It is urgent to explore the damage evolution of PBHFCC under chloride erosion and dry–wet cycle.

In this paper, polypropylene fiber and basalt fiber were composed of hybrid fibers. After several trials, the hybrid ratio of polypropylene fiber:basalt fiber = 1:1 was selected. Four kinds of ceramsite concrete with hybrid fiber volume content were prepared. The changes in appearance, mass damage, relative dynamic elastic modulus, compressive strength and splitting tensile strength of each group of PBHFCC under a chloride dry–wet cycle were measured, and the damage model of PBHFCC under a chloride dry–wet cycle was established with the change in relative dynamic elastic modulus as the index. The research results can provide a reference for the engineering application of PBHFCC in coastal areas.

## 2. Materials and Methods

### 2.1. Raw Material

(1)Cement: Huaxin brand P.O.42.5 ordinary Portland cement. The technical indicators are shown in Table 1.(2)Coarse aggregate: Yichang Guangda brand 900-grade high-strength gravel–shale ceramsite, performance parameters are shown in Table 2. The ceramsite is prewetted before preparing the concrete specimens.(3)Fine aggregate: ordinary river sand (medium sand), apparent density is 2.54 g/cm^3^, fineness modulus is 2.91.(4)Fiber: polypropylene fiber and basalt fiber. The performance parameters are shown in Table 3, and the photos are shown in Figure 1.(5)Water-reducing agent: HSC polycarboxylate high-performance water-reducing agent, the solid content of the water-reducing agent was 20%.(6)Water: tap water.

### 2.2. Mix Proportion of PBHFCC

The volume addition rate of polypropylene fiber is 0.1 vol %, and the volume addition rate of basalt fiber is 0 vol %, 0.05 vol %, 0.1 vol % and 0.15 vol %, respectively. Four kinds of mix proportion specimens were made for pre-experiment, and the cube compressive strength, splitting tensile strength and prism axial compressive strength of the four kinds of mix proportion specimens were tested. The results showed that when the volume addition rate of polypropylene fiber and basalt fiber is 0.1 vol %, the above three kinds of mechanical strength reach the maximum. Therefore, polypropylene fiber and basalt fiber were mixed at a ratio of 1:1. Three kinds of PBHFCC of 0.10 vol %, 0.15 vol % and 0.20 vol % were used for test-mixing by the absolute volume method. According to the 28 d compressive strength and 28 d splitting tensile strength measured by each test group, the optimal mix ratio was selected for the test. The specific mix ratio parameters are shown in Table 4. When the hybrid fiber content is 0.15 vol %, the compressive strength and splitting tensile strength are significantly improved compared with the reference group, followed by the 0.10 vol % group. When the content is 0.20 vol %, it is lower than the reference group by 1.87% and 2.73%, respectively. This shows that the group of hybrid fibers is added in too high an amount, and the negative effect of hybrid fibers appears. The strength test results are consistent with the literature [15].

### 2.3. The Test Scheme of Chloride Erosion and Dry–Wet Cycle

The chloride corrosion test of PBHFCC was carried out under the condition of a dry–wet cycle. A 3.5 vol % NaCl solution was used as an erosion solution. Referring to the dry–wet cycle test in the References [18,29,31], the dry–wet cycle was carried out by natural immersion and natural air-drying interaction. The specimens were immersed in the erosion solution for 3 d and then dried at room temperature for 5 d. The 8 d time point was a dry–wet cycle. After every 3 dry–wet cycles, the specimens were taken out to measure the relevant performance. The evaluation criteria of specimens’ deterioration include: (1) mass loss rate; (2) relative dynamic elastic modulus; (3) compressive strength; (4) splitting tensile strength.

The grouping of specimens is shown in Table 5. The schematic diagram of the specimen production is shown in Figure 2. A total of 264 pieces of 100 mm × 100 mm × 100 mm cube specimens (used for compressive strength and splitting strength test) and 12 pieces of 100 mm × 100 mm × 400 mm prism specimens (used for appearance damage observation, mass loss rate and relative dynamic elastic modulus test) were poured. The compressive strength and splitting tensile strength were measured according to Chinese standard GB/T 50081-2019 [32] (Standard for test methods of concrete physical and mechanical properties). The test was carried out on the YES-2000B digital (Jinan, China) display pressure testing machine. After 28 d of curing, the mechanical test was carried out. The test results were taken as the average of three specimens in each group.

The relative dynamic elastic modulus is one of the important indexes of the mechanical properties of materials, which can reflect the response ability and deformation degree of concrete under external loading. The model established by using the dimensionless index of the relative dynamic elastic modulus can predict and evaluate the overall mechanical properties of PBHFCC, so as to better understand the behavior and performance of PBHFCC.

## 3. Results and Discussion

### 3.1. Appearance Change of Test Piece

The appearance of the test piece at 0 d, 120 d (15 cycles) and 240 d (30 cycles) under the interaction of dry–wet cycles and 3.5% NaCl solution corrosion are, respectively, shown in Figure 3, Figure 4 and Figure 5. It can be seen from the figures that the surface of each group of specimens before the test was flat without obvious defects. With the increase in test time, the appearance damage of the specimens increased continuously. After 120 d of chloride erosion and dry–wet cycle, the area of microvoids on the surface of the four groups of specimens increased, and small cracks were produced. Among them, the crack distribution area of the GC-3 group (volume fraction of hybrid fiber = 0.20 vol %) is the largest, and the apparent damage degree is even slightly greater than that of the GC-0 group (volume fraction of hybrid fiber = 0.00 vol %). After 240 d, the surface damage of each group of specimens was significantly aggravated, and the degree and distribution area of cracks were obviously increased. Among them, the surface damage of the GC-2 group (volume fraction of hybrid fiber = 0.15 vol %) was relatively less, and the degree of deterioration was relatively light.

### 3.2. Change Rule of Mass Loss

The mass loss rate of PBHFCC specimens under the combined action of dry–wet cycle and 3.5 vol % NaCl solution is shown in Figure 6. Under the combined action of dry–wet cycle and chloride solution, with the increase in erosion age, the quality of four groups of PBHFCC specimens increased gradually in the first 72 d of erosion age and began to decrease rapidly after 72 d. Among them, the mass loss rate of the GC-2 group (volume fraction of hybrid fiber = 0.15 vol %) was the smallest in each stage of the test, and the mass loss rate was only 0.46% at 240 d erosion age, which indicated that this group of specimens was the least damaged by chloride salt erosion. In contrast, the mass loss rates of specimens in the GC-0, GC-1 and GC-3 groups were 0.54%, 0.50% and 0.56%, respectively, at 240 d erosion age, that is, the mass loss rate of specimens in the GC-3 group (volume fraction of hybrid fiber = 0.20 vol %) was the largest.

### 3.3. Change Rule of Relative Dynamic Elastic Modulus

The relative dynamic elastic modulus of PBHFCC specimens under the combined action of dry–wet cycle and 3.5 vol % NaCl solution is shown in Figure 7. The relative dynamic elastic modulus of the four groups of specimens increased first and then decreased with the increase in erosion age. The relative dynamic elastic modulus increased gradually during the first 48 d of erosion and decreased gradually after 48 d. During the whole erosion age, the relative dynamic elastic modulus of the GC-2 group was greater than that of the other three groups of specimens with the same erosion age. After 240 d of erosion, the relative dynamic elastic modulus of the GC-2 group decreased the least, with a value of 75.26%, followed by the GC-1 group, with a value of 73.93%. In contrast, the relative dynamic elastic modulus of the GC-0 group and the GC-3 group decreased the most, which were 71.85% and 69.37%, respectively. From Figure 3 to Figure 5 and Figure 7, it can be seen that the relative dynamic elastic modulus of concrete is related to the compactness of concrete in the four groups of PBHFCC specimens. The addition of an appropriate amount of hybrid fiber in concrete can effectively densify the pore structure of concrete and delay the occurrence of cracks [14]. Therefore, the relative dynamic elastic modulus loss of the GC-2 group (volume fraction of hybrid fiber = 0.15 vol %) is the smallest. However, in the GC-3 group (volume fraction of hybrid fiber = 0.20 vol %), the amount of hybrid fiber was too large, and the negative effect of hybrid fiber appeared, resulting in a large loss of relative dynamic elastic modulus.

### 3.4. Change and Analysis of Compressive Strength

The compressive strength of PBHFCC specimens under the combined action of dry–wet cycle and 3.5 vol % NaCl solution is shown in Figure 8. It was taken as the average of three specimens in each group. The standard deviation is between 1.77 and 3.32. The compressive strength of the four groups of PBHFCC increased before 72 d and began to decrease after 72 d. At each durability damage age, the compressive strength of the GC-2 group was the largest, followed by GC-1, GC-0 and GC-3. This phenomenon is well related to the change rule of specimens quality in Figure 6. The compressive strength of the four groups of specimens reached the maximum value at the 72 d erosion age, and the compressive strength of the GC-2 group reached the maximum value of 66.2 MPa. Compared with the specimens without hybrid fibers, it was increased by 12.2%. Followed by GC-1, GC-0 and GC-3, the maximum compressive strength reached 63.2 MPa, 59.0 MPa and 56.1 MPa, respectively. At 240 d erosion age, the compressive strength loss of the GC-2 group was the smallest, which was 43.8 MPa. The compressive strengths of GC-1, GC-0 and GC-3 were 37.7 MPa, 34.5 MPa and 30.5 MPa, respectively.

It can be seen from the above analysis that the addition of hybrid fibers of 0.15 vol % had the most obvious effect on improving the compressive strength of ceramsite concrete, followed by the addition of hybrid fibers of 0.10 vol %. The negative effect of hybrid fiber appears in the hybrid fiber specimens with 0.20 vol %, which makes the compressive strength of concrete lose more, and the compressive strength is lower than that of the specimens without fiber.

### 3.5. Change and Analysis of Splitting Tensile Strength

The splitting tensile strength of PBHFCC specimens under the combined action of dry–wet cycle and 3.5 vol % NaCl solution is shown in Figure 9. It was taken as the average of three specimens in each group. The standard deviation is between 0.18 and 0.35. Under the combined action of dry–wet cycle and chloride erosion, the change rule of splitting tensile strength of hybrid fiber ceramsite concrete specimens is similar to that of compressive strength. The splitting tensile strength of each group of specimens reached the peak at 72 d, gradually increased in the first 72 d, and decreased rapidly after 72 d. Among the four groups of specimens, the splitting strength of the GC-2 group was the highest. The maximum splitting strength of the 72 d specimens reached 5.16 MPa, which was 4.45% higher than that of the uncorroded specimens. Compared with the specimens without hybrid fibers, it was increased by 14.2%. After 240 d, it decreased to 4.25 MPa, and the loss was 13.97% compared with the strength of uncorroded specimens. The splitting strength of the GC-3 group was the lowest, which increased to the maximum value of 4.46 MPa at 72 d, and the strength increased by 7.73% compared with the uncorroded specimens. After 240 d, it decreased to 3.28 MPa, and the loss was 20.77% compared with the uncorroded specimens. It can be seen that after adding an appropriate amount of hybrid fiber to ceramsite concrete, the splitting tensile strength of the specimens changes more stably and is less affected by dry–wet cycle and chloride salt erosion, while without adding hybrid fiber and too much hybrid fiber content will lead to more loss of splitting tensile strength.

### 3.6. Analysis of Damage Causes

The diffusion of chloride ions in concrete can be described by Fick’s law [33]. In the early stage of chloride erosion, Cl^−^ reacts with C_3_A in concrete to form expansive crystalline products such as Friedel’s salt [34]. Therefore, the quality is on the rise in the early stage of erosion. Expansive products not only fill the internal pores of concrete, but also increase the bonding force between concrete and hybrid fibers, and reduce the thickness of the concrete interface and water film layer of hybrid fibers. The hybrid fiber has good crack resistance and tensile properties. The fibers are located and distributed across the cracks that induce bridging action, in which they effectively initiate restraining the propagation of microcracks. The quality of the specimen is increased by the above comprehensive effects, and the mechanical properties are also improved (including relative dynamic elastic modulus, compressive strength, splitting strength).

With the increase in erosion age, the crystallization products accumulate and fill the pores, and the expansion pressure increases gradually. Under the acceleration of dry–wet cycles, when the expansion pressure is greater than the ultimate tensile stress of concrete, a small amount of slurry begins to peel off and microcracks appear on the surface of the concrete. This makes the concrete produce crack damage, which causes the specimen quality, relative dynamic elastic modulus, compressive strength, splitting strength and other indicators to decline rapidly in the later stage.

Adding an appropriate proportion of hybrid fibers can effectively improve the pore structure of concrete, effectively block the generation of concrete cracks, and delay the damage of specimens. However, when the amount of hybrid fiber is too large, the negative effect of hybrid fiber will appear, which will deteriorate the performance of concrete and accelerate the damage to the specimen. This is similar to that reported in the literature [30], i.e., when the volume content of basalt fiber is 0.15%, the maximum value is reached. The addition of basalt fiber weakens the ability of concrete to resist chloride-ion penetration. Zheng [35] also reports that according to the microstructure analysis data, an appropriate amount of fibers can make the internal structure of the fiber dispersion more uniform and with moderate spacing, resulting in minimum porosity. With the excessive addition of fibers, the fibers begin to cluster together, which reduces the strength of the concrete.

### 3.7. Damage Model of PBHFCC under Chloride Erosion and Dry–Wet Cycle

After the concrete specimen is damaged by erosion, microcracks will be generated inside the concrete, and the phenomenon of mortar and concrete matrix separation will occur. The relative dynamic elastic modulus can accurately reflect the internal damage degree of concrete. Therefore, the relative dynamic elastic modulus value is selected as the damage variable, which can represent the damage degree of PBHFCC with different dosages under the action of erosion.

After the interaction of 3.5% NaCl solution and dry–wet cycle, the change in relative dynamic modulus of PBHFCC can be divided into two stages: (1) Rising stage, that is, the process of concrete pore saturation; (2) Decline stage, that is, the process of concrete pore supersaturation. The critical point of the two stages is the time point when the internal pressure of the concrete pore equals the ultimate tensile strength of concrete, which is set as t_12_.

Since the first stage is the rising stage, a linear function (E_rd1_ = At_1_ + B) is used to express the change process of relative dynamic modulus of elasticity. When t_1_ = 0, the specimen has not been damaged, E_rd1_ = 100, so B = 100. Because the second stage is the descending stage, based on the process of damage degradation of concrete having uniform acceleration, the loss process of relative dynamic modulus of elasticity in the second stage is represented by quadratic polynomial (E_rd2_ = Ct_2_^2^ + Dt_2_ + E), that is, the whole damage degradation process of concrete corroded by chloride under the dry–wet cycle is
(1)$$E_{rd}=f(t)=\left\{\begin{array}{l} E_{rd1}=f(t_{1})=At_{1}+100\\ E_{rd2}=f(t_{2})=C{t_{2}}^{2}+Dt_{2}+E \end{array}\right.$$

Among them: A, C, D, E are test parameters. A represents the initial speed of concrete damage deterioration in the first stage, and D represents the initial speed of concrete damage in the second stage. A and D are proportional to the erosion rate of concrete by salt ions. The greater A and D, the faster the erosion rate of concrete. t is the erosion time (d), t_1_ ≤ t_12_ ≤ t_2_, t_12_ is the transition point from the first stage to the second stage.

Equation (1) satisfies the following boundary conditions:When t = 0, E_rd1_ = f (0) = 100;n the whole function interval t ∈ [0, t], f(t) ≥ 0;When t_1_ = t_2_ = t_12_, f(t_1_) = f(t_2_) = f(t_12_);When 0 ≤ t ≤ t_12_, f′(t) > 0; When t_12_ ≤ t_2_ ≤ t_max_, f″(t_2_) < 0.

Among them, f′(t) represents the speed of concrete damage deterioration, and f″(t) represents the acceleration of concrete damage deterioration.

Derivation of E_rd_:(2)$$f^{\prime }(t)=\frac{dE_{rd}}{dt}=\left\{\begin{array}{l} f^{\prime }(t_{1})=A\\ f^{\prime }(t_{2})=2Ct_{2}+D \end{array}\right.$$

The equation is derived twice:(3)$$f^{\prime \prime }(t)=\frac{d^{2}E_{rd}}{dt^{2}}=\left\{\begin{array}{l} f^{\prime \prime }(t_{1})=0\\ f^{\prime \prime }(t_{2})=2C \end{array}\right.$$

Among them, 2C represents the damage acceleration of concrete after entering the second stage, that is, the concrete accelerates the damage uniformly with the initial speed of D and the acceleration of 2C. The greater the 2C, the faster the damage rate, and the more serious the damage and deterioration of concrete.

The data of relative dynamic elastic modulus are substituted into Equation (1) for fitting, and the fitting results are shown in Figure 10 and Table 6.

Among the four groups of specimens, t_12_ of the GC-2 group is the smallest, followed by the GC-1 group, and t12 of the GC-0 group is the largest. This is because the addition of hybrid fibers can reduce the pores of concrete. The expansion crystallization products generated by the specimens during the dry–wet cycle are faster than the specimens without hybrid fibers at filling the pores of concrete, which makes the damage and deterioration of PBHFCC enter the second stage faster.

In order to analyze the influence of mixed fiber content on the chloride corrosion resistance of concrete, the influence coefficient K of mixed fiber is introduced. It can be seen from the table that the coefficients of the equation increase first, then decrease and then increase with the increase in fibers. Therefore, the cubic polynomial is used to characterize the effect of the volume of mixed fibers in chloride environment on the damage evolution equation of concrete.

Set
k = a + bx + cX^2^ + dX^3^(4)

In the equation, K is the influence coefficient of the volume content of hybrid fiber on the damage evolution equation of concrete, X is the content of hybrid fiber, a, b, c, d are the test-fitting constants. The influence factors are as follows.

Coefficient A: k = 0.17927 + 0.29976X + 21.56864X^2^ − 105.2403X^3^

Coefficient C: k = 3.55004E − 4 + 0.01487X − 0.20436X^2^ − 0.20436X^3^

Coefficient D: k = −0.31113 − 4.79672X + 66.71511X^2^ − 221.0406X^3^

Coefficient E: k = 125.44314 + 210.72335X − 2415.70692X^2^ + 7379.3544X^3^

Therefore, the damage evolution equation of concrete considering the volume content of hybrid fiber is as follows:(5)$$E_{rd}=f\left(t\right)=\left\{\begin{array}{l} E_{rd1}=f(t_{1})=\left(0.17927+0.29976X+21.56864X^{2}-105.2403X^{3}\right)t_{1}+100\\ E_{rd2}=f(t_{2})=(3.55004E-4+0.01487X-0.20436X^{2}-0.20436X^{3}){t_{2}}^{2}\\ +(-0.31113-4.79672X+66.715X^{2}-221.0406X^{3})t^{2}\\ +\left(125.44314+210.72335X-2415.70692X^{2}+7379.3544X^{3}\right) \end{array}\right.$$

## 4. Conclusions

In this paper, through the dry–wet cycle–chlorine salt erosion experiment of PBHFCC with different volume fiber content, the apparent damage degree, mass loss rate, relative dynamic elastic modulus, compressive strength and splitting tensile strength of PBHFCC are analyzed. The main conclusions are as follows:(1)The durability of PBHFCC is affected by the number of erosion d and fiber content. The measured mechanical indexes increased first and then decreased with the increase in action time. An appropriate amount of fiber incorporation (0.10 vol %, 0.15 vol %) can effectively improve the durability of ceramsite concrete, while too-high fiber incorporation (0.20 vol %) will produce negative effects of hybrid fibers and aggravate erosion damage.(2)The apparent damage degree of each group of specimens increased gradually with the increase in action time. The durability damage of the specimens with hybrid fiber of 0.20 vol % was the most serious, and the damage degree was even slightly larger than that of the specimens without hybrid fiber. The damage degree of the specimen with hybrid fiber of 0.15 vol % was lighter.(3)The mass of PBHFCC in each group increased gradually during the first 72 d of erosion, and decreased rapidly after 72 d. The mass loss rate of the specimens with hybrid fiber of 0.15 vol % was the smallest in each stage of the test.(4)The relative dynamic elastic modulus increased gradually during the first 48 d of erosion and decreased gradually after 48 d. The relative dynamic elastic modulus of the specimen with hybrid fiber of 0.15 vol % was larger than that of the other three groups of specimens with the same erosion age.(5)The compressive strength of PBHFCC in each group increased gradually in the first 72 d and then decreased gradually. When the erosion age is 72 d, the compressive strength and splitting tensile strength of the hybrid fiber specimen with 0.15 vol % volume content reach the peak, which are 66.2 MPa and 5.16 MPa, respectively. Compared with the specimens without hybrid fibers, they were increased by 12.2% and 14.2%, respectively.(6)Taking the relative dynamic elastic modulus value as the damage variable, the linear function is used to represent the rising stage in the early stage, and the quadratic polynomial is used to represent the falling stage in the later stage. The damage evolution equation of PBHFCC considering the volume content of hybrid fiber under the action of chloride erosion and dry–wet cycle is well fitted, which can provide reference for the engineering design of PBHFCC in coastal areas.

## Figures and Tables

**Figure 1 polymers-15-04179-f001:**
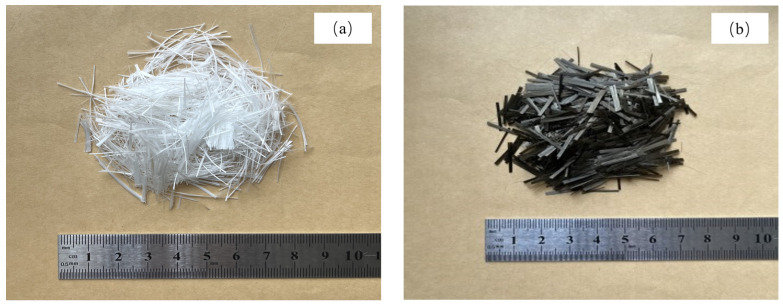
Fiber: (**a**) polypropylene fiber, (**b**) basalt fiber.

**Figure 2 polymers-15-04179-f002:**
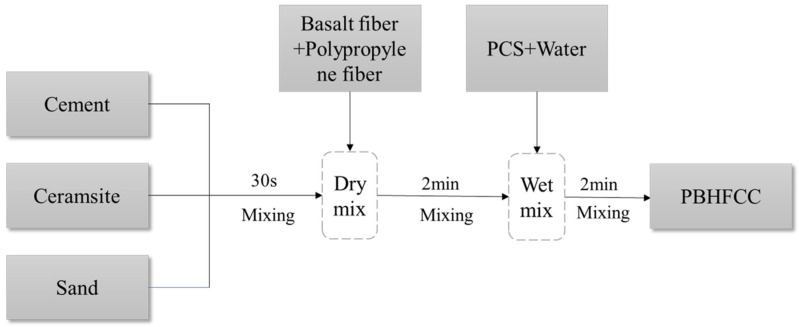
The schematic diagram of the specimen production.

**Figure 3 polymers-15-04179-f003:**
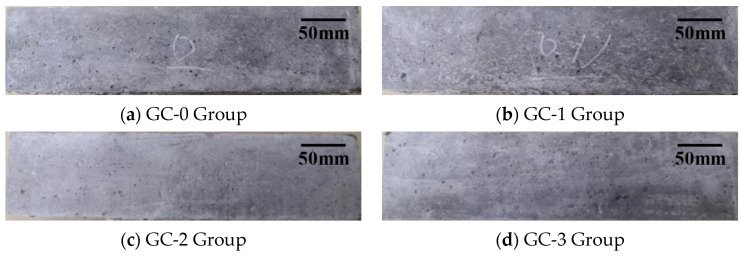
Appearance of test piece under the interaction of dry–wet cycles and 3.5% NaCl solution corrosion after 0 d.

**Figure 4 polymers-15-04179-f004:**
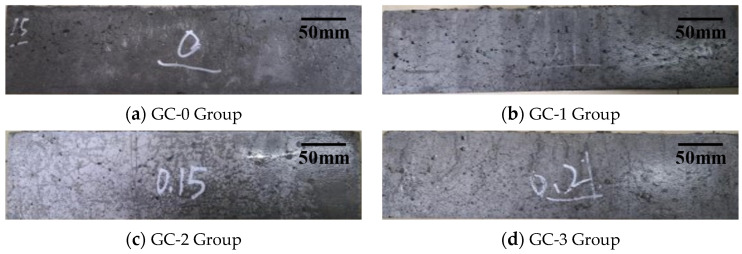
Appearance of test piece under the interaction of dry–wet cycles and 3.5% NaCl solution corrosion after 120 d.

**Figure 5 polymers-15-04179-f005:**
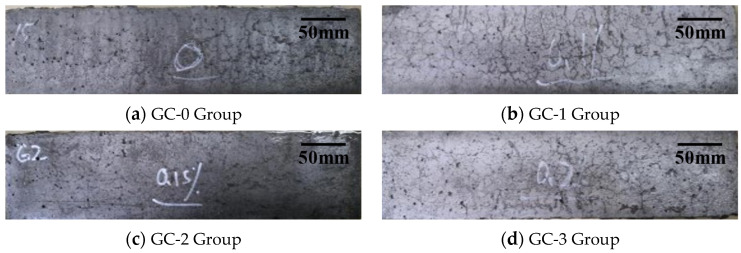
Appearance of test piece under the interaction of dry–wet cycles and 3.5% NaCl solution corrosion after 240 d.

**Figure 6 polymers-15-04179-f006:**
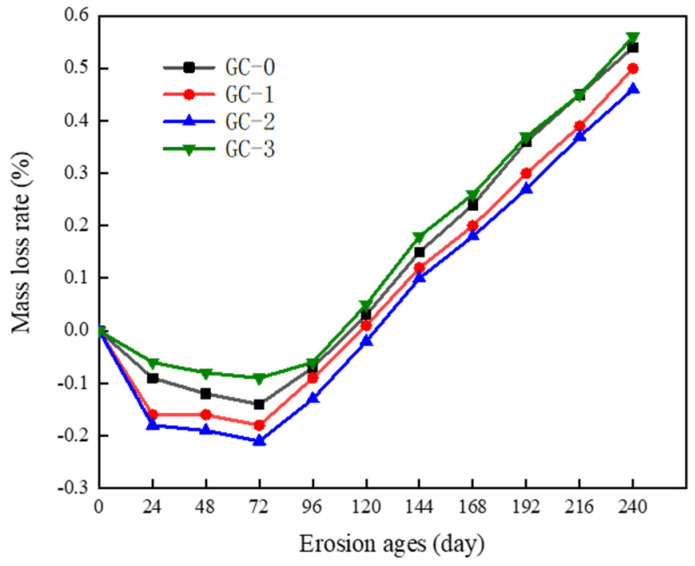
Change in mass loss rate of specimens corroded by dry–wet cycle in 3.5% NaCl solution.

**Figure 7 polymers-15-04179-f007:**
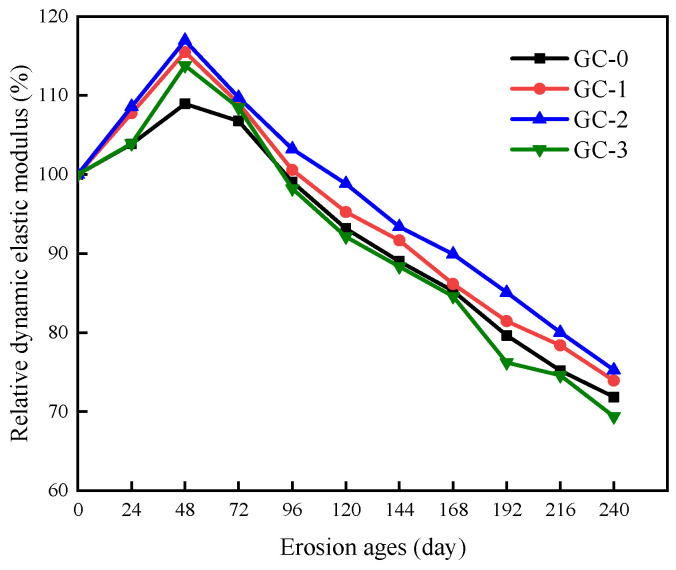
Change in relative dynamic elastic modulus of specimens corroded by dry–wet cycle in 3.5% NaCl solution.

**Figure 8 polymers-15-04179-f008:**
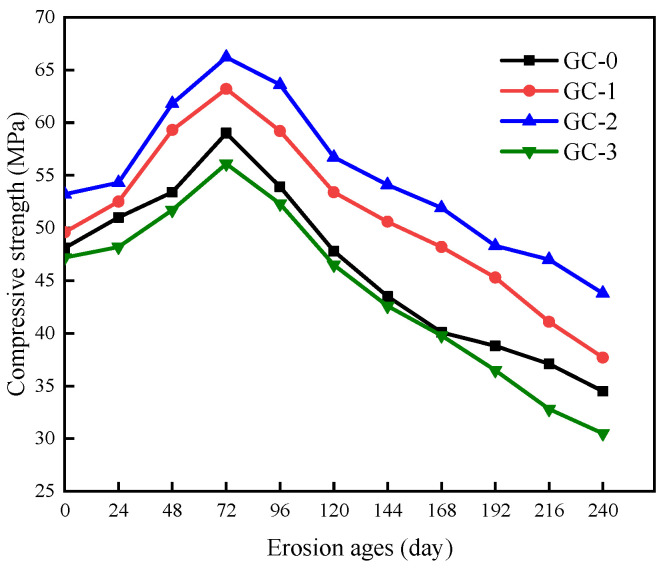
Change in compressive strength of specimens corroded by dry–wet cycle in 3.5% NaCl solution.

**Figure 9 polymers-15-04179-f009:**
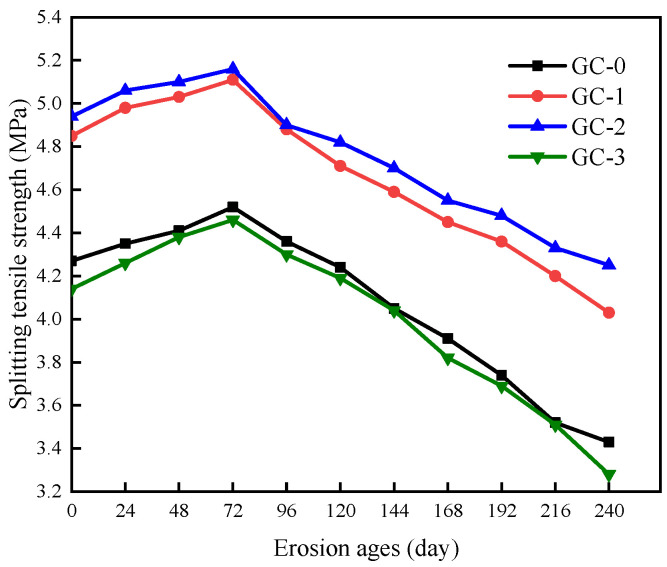
Change in splitting tensile strength of specimens corroded by dry–wet cycle in 3.5% NaCl solution.

**Figure 10 polymers-15-04179-f010:**
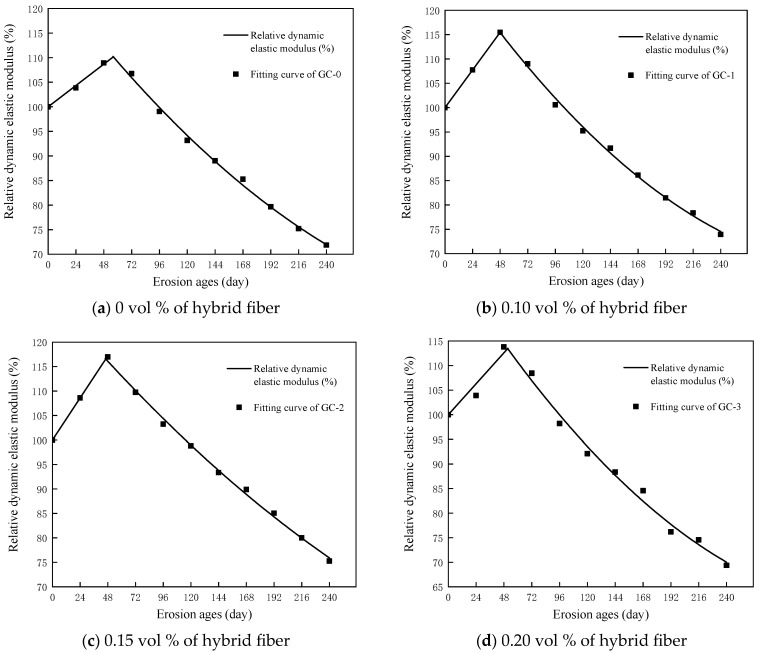
The damage evolution law of relative dynamic elastic modulus of specimens corroded by dry–wet cycle in 3.5% NaCl solution.

**Table 1 polymers-15-04179-t001:** Technical indicators of physical properties related to cement.

Performance Index	Stability	Initial Setting Time (Min)	Final Setting Time (Min)	Compressive Strength (MPa)		Flexural Strength (MPa)	
				3 d	28 d	3 d	28 d
Measured value	Qualification	110	350	22.5	47.3	4.8	7.6

**Table 2 polymers-15-04179-t002:** Aggregate performance parameters.

Materials Name	Grain Size (mm)	Packing Density (kg/m^3^)	Apparent Density (kg/m^3^)	Compressive Strength of Concrete Cylinder (MPa)	2 h Water Absorption (%)	Fineness Modulus	Mud Content (%)
Ceramsite	5~20	815	1505	6.2	2.46	-	-
Sand	-	1560	2650	-	-	≤2	2.65

**Table 3 polymers-15-04179-t003:** Fiber performance parameters.

Fiber Name	Shape	Length (mm)	Diameter (um)	Density (g/cm^3^)	Elastic Modulus (GPa)	Extension at Break (%)	Tensile Strength (MPa)
Polypropylene fiber	Bunchy monofilament	18	10~20	0.91	≥3.6	6~20	≥400
Basalt fiber	Bunchy monofilament	12	20~30	2.65	95~110	2.5~3.5	3800~4900

**Table 4 polymers-15-04179-t004:** Mix proportion and strength parameters of PBHFCC.

Group No.	Volume Fraction of Hybrid Fiber (%)	Cement (kg/m^3^)	Ceramsite (kg/m^3^)	Sand (kg/m^3^)	Water (kg/m^3^)	PCS (kg/m^3^)	Average Compressive Strength of 28 d (MPa)	Average Splitting Tensile Strength of 28 d (MPa)
Reference group (GC-0)	0.00	540	554	730	152	5.4	48.1	4.27
Control group 1 (GC-1)	0.10	540	554	730	152	5.4	49.6	4.85
Control group 2 (GC-2)	0.15	540	554	730	152	5.4	53.2	4.94
Control group 3 (GC-3)	0.20	540	554	730	152	5.4	47.2	4.14

**Table 5 polymers-15-04179-t005:** Grouping of dry–wet cycle and salt solution interaction test.

Group No.	Solution Type and Volume Concentration	Volume Fraction of Hybrid Fiber (%)	Number of Compression Strength Test Pieces (Piece)	Number of Splitting Tensile Strength Test Pieces (Piece)	Mass Loss Rate/Relative Dynamic Modulus of Elasticity Test Pieces (Piece)
GC-0	NaCl 3.5% (in volume percent)	0.00	3 × 11	3 × 11	3
GC-1		0.10	3 × 11	3 × 11	3
GC-2		0.15	3 × 11	3 × 11	3
GC-3		0.20	3 × 11	3 × 11	3

**Table 6 polymers-15-04179-t006:** E_rd_ damage-fitting coefficient of specimens corroded by dry–wet cycle in 3.5% NaCl solution.

Group	Fit Parameter Value					Fitting Accuracy R^2^
	A	C	D	E	t_12_	
GC-0	0.18108	3.5859 × 10^−4^	−0.31427	126.71024	55.937	0.997
GC-1	0.32292	4.69783 × 10^−4^	−0.34817	131.04824	47.345	0.996
GC-2	0.35792	2.40124 × 10^−4^	−0.27834	128.89248	45.763	0.996
GC-3	0.26267	4.93862 × 10^−4^	−0.37393	131.30744	51.451	0.982

## Data Availability

The data presented in this study are available on request from the corresponding author.

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
