# Peer review of "Damage Evolution of Polypropylene–Basalt Hybrid Fiber Ceramsite Concrete under Chloride Erosion and Dry–Wet Cycle"

_polymers, 2023, doi:10.3390/polym15204179_

Round 1
Reviewer 1 Report
The manuscript, entitled Damage Evolution of Polypropylene-Basalt Hybrid Fiber Ceramsite Concrete under Chloride Erosion and Dry-Wet Cycle," presents an experimental study conducted on the obtaining of concrete with a mix of 1 to 1 polypropylene and basalt. However, the literature overview is vague, the novelty of the study wasn’t presented, the future directions and limitations section is missing, and other issues must be addressed. The paper needs major revisions before it is processed further. Some comments follow:
Abstract: Please highlight the novelty of the paper in the first part of the abstract.
This section must be suitable for separate presentations (independent of the manuscript text body); therefore, it should include novelty, materials and methods, and results presented in quantitative evaluation.
Introduction Section
The introduction should be significantly improved. Some citations were introduced in bulk format ([2-5], [17-19], etc.). Please distribute the citations in the text depending on the information they confirm. For example, in the sentence "improve the strength, toughness, crack resistance, and freeze-thaw resistance of ceramsite concrete," Citations should be introduced for each individual property or characteristic.
Also, please present relevant literature in quantitative form. Include clear values for the tested properties and the effect of key parameters on those properties.
The authors stated that "polypropylene fiber and basalt fiber"; therefore, what is the novelty of this study? Please clearly highlight what makes the results from this research different from those previously reported and how this paper will extend the knowledge in this field.
Materials and methods
Please describe each piece of equipment that has been used (type and manufacturer) and the parameters that influence the obtained results.
Please provide a graphical procedure for obtaining the samples.
Results and discussions
Please introduce a scale bar on each part of figures 1 to 4.
How many samples have been tested from each batch? The repeatability and credibility of the results are strongly related to the number of tests performed. Each test should be performed on a batch of samples (since this is a study conducted on materials made of mixtures), and the mean values should be used to create the graphs or plots (while the deviation values should be provided for each point).
A modeling part of the results should be included to extend the results obtained in this study; otherwise, the results are only limited to the very small number of mixtures that have been tested.
Also, the authors stated that a ratio of 1 to 1 basalt to polypropylene fibers was selected after a preliminary trial, but the motivation is missing (what will happen if the ratio is different?).
The obtained material should be characterized for all relevant characteristics. Please analyze the porosity and pore size distribution, the microstructure of the damaged samples, thermal behavior, failure mechanism, etc.
Discussion section. The discussion section is missing. In the discussion section, a clear correspondence and comparison between the results of this study and those from the literature should be provided. Please improve. Currently, the discussion section includes some comments and appreciation about the obtained results without any comparison with the literature.
Conclusion
Please improve the quality of the conclusions and present them in a more comprehensive manner (currently, the conclusion section is way too long and vague).
Future directions and limitations: Please provide some future directions and limitations of the study. This section is very important for studies that propose materials with industrial applications.
Reviewer 2 Report
Dear Authors,
Thank you for you manuscript, here will be following remarks and comments:
1. Please shorten your abstract to 200 words and make it less general – more detailed on what your main research outcome and novelty;
2. Introduction should be split in more paragraphs with a certain structure; It is quite general and hectic; Include table or chart on the previous research;
3. Materials & methods section should include all information on the test methods and equipment used; description of samples, reference to standards;
4. Reorganize and compact your tables 2 & 3 & 4; difficult to read; should be one very compact table; mix design should include volume units to have all components with their weigths in 1m3.
5. Figure 1, should have scale;
6. Figure 2, 3 & 4 doesn’t seems to have the same marking over time; Please include photos of the whole samples for overview;
7. Please provide overview of the amount of samples tested; include stdev in your results;
8. 3.7 should be included in Methodology;
9. Conclusions must be elaborated and written shortly with main outcome (too general and broad).
10. Novelty?
11. The earliest reference is dated to 2017! What about research before that time?
ok
Round 2
Reviewer 1 Report
Dear Authors,
You have done a great job in revising the paper.
Best regards,
Reviewer 2 Report
No further comments
Ok